# Prognostic Biomarkers of Salvage Chemotherapy Following Nivolumab Treatment for Recurrent and/or Metastatic Head and Neck Squamous Cell Carcinoma

**DOI:** 10.3390/cancers12082299

**Published:** 2020-08-15

**Authors:** Takahiro Wakasaki, Ryuji Yasumatsu, Muneyuki Masuda, Toranoshin Takeuchi, Tomomi Manako, Mioko Matsuo, Rina Jiromaru, Ryutaro Uchi, Noritaka Komune, Teppei Noda, Takashi Nakagawa

**Affiliations:** 1Department of Otorhinolaryngology, Graduate School of Medical Sciences, Kyushu University, Fukuoka 812-8582, Japan; yasuryuj@qent.med.kyushu-u.ac.jp (R.Y.); matsumio@qent.med.kyushu-u.ac.jp (M.M.); rjiro@med.kyushu-u.ac.jp (R.J.); urentcello@gmail.com (R.U.); norikomu007@gmail.com (N.K.); teppei@dev.med.kyushu-u.ac.jp (T.N.); nakataka@qent.med.kyushu-u.ac.jp (T.N.); 2Department of Head and Neck Surgery, National Hospital Organization, Kyushu Cancer Center, Fukuoka 812-8582, Japan; masuda.muneyuki.pg@mail.hosp.go.jp; 3Department of Otorhinolaryngology, Kitakyushu Municipal Medical Center, Kitakyushu 802-8561, Japan; ayatora15@hotmail.co.jp (T.T.); splv_tomato_85144@yahoo.co.jp (T.M.)

**Keywords:** biomarker, chemotherapy, C-reactive protein, distant metastasis, head and neck squamous cell carcinoma, neutrophil to lymphocyte ratio, nivolumab, recurrent

## Abstract

Recent studies have suggested the benefit of salvage chemotherapy (SCT) after immune checkpoint inhibitor (ICI) treatment for recurrent and metastatic head and neck squamous cell carcinoma (R/M HNSCC). We retrospectively examined the outcome of SCT and the usefulness of the serum C-reactive protein level (CRP) and neutrophil-to-lymphocyte ratio (NLR) as prognostic biomarkers. Thirty-nine patients with R/M HNSCC were enrolled in this study. Twenty-five patients (64.1%) received combination chemotherapy of weekly paclitaxel and cetuximab (PC) as SCT, and 14 patients (35.9%) received tegafur-gimestat-otastat potassium (S1), an oral fluoropyrimidine. In all patients, the response rate, disease control rate, median progression-free survival (PFS), and median overall survival (OS) were 45.2%, 85.7%, 6.5 months, and 13.5 months, respectively. No chemotherapy-related deaths were observed. These PC groups had low CRP (<1.2 mg/dL) or low NLR (<7.0) values at the time of SCT induction, which was significantly associated with an improved OS (*p* = 0.0440, *p* = 0.0354). A multivariate analysis also showed that a lower CRP value was significantly associated with a better OS (*p* = 0.0078). We clarified the usefulness of the PC and S1 regimens as SCT. In addition, SCT with the PC regimen showed a better prognosis with a lower CRP or NLR at induction than a higher CRP or NLR. This is the first report on biomarkers of SCT in R/M HNSCC.

## 1. Introduction

Recently, a paradigm shift in the treatment of recurrent or metastatic head and neck squamous cell carcinoma (R/M HNSCC) has occurred. Palliative chemotherapy for patients with R/M HNSCC has led to marked changes in outcomes since the EXTREME trial in 2008 and the CheckMate 141 trial in 2016 [1,2]. Especially, with regard to immune checkpoint inhibitors (ICIs), long-term survivors have emerged whose performance status is maintained [2], although the number of patients who respond to nivolumab alone is somewhat limited (approximately 20%). For patients with progressive disease status after ICI treatment, another therapy sometimes could be administered to extend their life. Several studies have reported that salvage chemotherapy (SCT) administered after immunotherapy would be effective for patients with metastatic non-small-cell lung cancer (NSCLC), metastatic melanoma, B cell lymphoma, advanced gastric cancer, and metastatic urothelial carcinoma [3,4,5,6,7,8,9]. Furthermore, SCT has been reported to be an effective treatment for HNSCC in R/M as well as other carcinomas and is attracting attention as a sequential treatment [10,11]. Currently, combination chemotherapy of weekly paclitaxel and cetuximab (PC) or oral tegafur-gimestat-otastat potassium (S1) is administered as SCT after ICI treatment in our institution and associated facilities. However, predicting the response to treatment with SCT as well as ICIs remains challenging. The long-term efficacy and prognosis vary greatly among patients.

Recently, cancer-related systemic inflammation has been shown to be a major determinant of disease progression and survival in most cancers [11,12,13]. A common inflammatory marker, C-reactive protein (CRP), and the neutrophil to lymphocyte ratio (NLR), etc. have been reported to be prognostic markers for palliative chemotherapy. Elevated NLR and/or elevated CRP have been associated with poor survival after chemotherapy in various cancers [12,13,14]. The effects of these markers on the prognosis are interesting and of great clinical significance, since there have been no studies on the biomarkers that predict the prognosis and response of SCT after ICI for R/M HNSCC.

Therefore, in the present study, we aimed to evaluate the clinical usefulness of the PC regimen and S1 as SCT following ICI administration for patients with R/M HNSCC. We also investigated whether or not CRP and NLR predict the therapeutic effects of PC regimen after ICI treatment in patients with R/M HNSCC.

## 2. Results

### 2.1. Patients Characteristics

From 1 April 2017 to 30 November 2019, 136 patients with R/M HNSCC were treated with nivolumab at Kyushu University Hospital, National Kyushu Cancer Center, and Kitakyushu Municipal Medical Center, Japan. All tumors were histologically confirmed to be squamous cell carcinoma. Twenty-two patients were continuing with ICI therapy because good responses were maintained. Sixty patients were selected to receive best supportive care (BSC) after progressive disease (PD) with ICI therapy. Eight patients became untraceable after ICI therapy. Finally, 46 patients were administered PC or S-1 as SCT after ICI therapy turned out to be PD and they were enrolled in this study (Figure 1). Seven patients were not included this study because they were before the first assessment of treatment efficacy after the start of SCT. Finally, 39 patients were included in this retrospective study, including the 10 patients we have previously reported as a case series [11]. The clinical characteristics of the 39 patients are summarized in Table 1. The patients included 32 men and 7 women (median age, 65 years; range, 33–78 years). The primary site of the tumor varied. In 25 patients (64.1%), the primary cancers were located in the oral cavity, oropharynx, hypopharynx, or larynx. All tumors were histologically confirmed to be squamous cell carcinoma (SCC). Twenty-seven patients (69.2%) had distant metastasis with or without locoregional recurrence, and 12 (30.8%) had a locoregional recurrence. The Eastern Cooperative Oncology Group performance status (PS) was 0–1 in 40 patients and 2 in two patients. They were treated with PC (25/39, 64.1%) or S1 (14/39, 35.9%) as SCT. The previous definitive therapies and the administration lines of SCT were summarized in Appendix A. The line of administration of SCT was defined as the line of administration as palliative chemotherapy, including ICI therapy. The rates of second-line, third line, and fourth-line treatment were 53.8% (21/39), 35.9% (14/39), and 7.7% (3/39), respectively. The previous palliative chemotherapies included platinum (16 of 39 patients (PC group, 8/25; S1 group, 8/14)), taxanes (5 of 39 patients (PC group, 2/25; S1 group, 3/14)) and cetuximab (17 of 39 patients (PC group, 8/25; S1 group, 9/14)). Thirty-one of 39 patients were judged as platinum-refractory before the administration of nivolumab, because we experienced cancer progression within 6 months after the initiation of platinum chemotherapy in the context of primary or recurrent disease. Eight patients were judged as platinum intolerant before the administration of nivolumab because of renal dysfunction, old age, or poor PS. Platinum refractory was defined as cancer progression within 6 months after the initiation of platinum chemotherapy in the context of primary or recurrent disease. In these 39 patients, the objective response rate (ORR; complete response (CR) + partial response (PR)) and disease control rate (DCR; CR + PR + stable disease (SD)) of nivolumab were 28.2% and 61.5%, respectively. The effectiveness of nivolumab in this group of patients was similar to that in previous reports on R/M HNSCC. Patient follow-up lasted until death or the cutoff date (29 February 2020). The median follow-up interval was 10.1 months (range 2.9–24.3 months) in all patients, and 10.9 months (range 4.4–23.4 months) in the PC patients.

### 2.2. Overall Treatment Efficacy of SCT

The ORR was 46.2% (18/39), and the DCR was 89.7% (35/39) in patients treated with chemotherapy after nivolumab treatment (Table 2). Furthermore, ORR in the patients treated with PC was significantly higher in comparison to patients treated with S1 therapy as SCT (15/25, 60%, *p* = 0.0428) (Table 2). DCR in the patients treated with PC was also high (24/25, 96%), however, there was no significant difference compared to S1. The Kaplan-Meier survival curves of the 39 patients are presented in Figure 2A,B. The estimated median overall survival (OS) and progression-free survival (PFS) from the first dose after SCT was 13.4 months and 6.5 months, respectively. There was no significant difference in the OS (Figure 2C) or the PFS (Figure 2D) between the patients administered PC and S1.

### 2.3. Adverse Events

All grade adverse events (AEs) and Grade 3/4 and above AEs during SCT were observed in 32 patients (82.1%) and 17 patients (43.6%), respectively (Appendix A). In the PC patients, all grade AEs and grade 3/4 AEs were significantly more frequent than in the S1 patients (*p* = 0.00120, *p* = 0.0449). All grade AEs and grade 3/4 during PC were observed in 24 patients (96.0%) and 16 patients (64.0%). All grade AEs included leukopenia in 16 patients (64.0%), neutropenia in 14 (56.0%), and skin disorder in 13 (52.0%). Grade 3/4 bone marrow suppression occurred in 15 patients (60.0%), Grade 3 skin disorder occurred in 5 (20.0%), and Grade 3/4 hypomagnesemia occurred in 4 (16.0%). All-grade interstitial pneumonia occurred in four cases (16.0%), including one case with grade 3 pneumonia (4.0%). In addition, one case of grade 3 endotracheal hemorrhaging and one case of grade 3 catheter-associated infection were noted. There were no chemotherapy-related deaths.

### 2.4. Association of Pre-SCT NLR or Pre-SCT CRP and the Prognosis in Patients Treated with PC

The median interval between the last dose of nivolumab and the first dose of SCT in all patients was 21 days (14–394 days). The Kaplan-Meier curves for each factor are presented in Figure 3A–D. Patients with low pre-SCT NLR and low pre-SCT CRP had a significantly better prognosis in terms of OS (*p* = 0.0232, *p* = 0.0301) (Figure 2B and Figure 3A), whereas there was no significant difference in PFS among two groups (*p* = 0.181, *p* = 0.468) (Figure 3C,D). The results of the univariate analysis of factors associated with OS and PFS in the PC patients are summarized in Table 3. A low CRP (<1.2 mg/dL) and NLR (<7.0) at the induction of SCT were significantly associated with a better OS (hazard ratio (HR): 0.324, HR: 0.354) (Table 3). Furthermore, the multivariate analysis revealed that low pre-SCT CRP was independent prognostic factors for OS (HR 0.0929) (Table 4).

## 3. Discussion

The National Comprehensive Cancer Network guideline describes nivolumab as a preferred regimen in cases with disease progression on or after platinum therapy. In our institutions, nivolumab is mainly administered for progressive disease after chemoradiotherapy (combined with platinum) or PFC therapy for R/M HNSCC. In the treatment of R/M HNSCC in the clinical setting, ICIs, including nivolumab, have been predominantly administered in early lines of treatment. The nivolumab treatment that has emerged in recent years includes many cases that can be treated while maintaining QOL. Thus, the role of SCT after nivolumab treatment is increasing. However, the drugs used for SCT include taxanes, fluoropyrimidine, platinum, and cetuximab, which are considered to be associated with a higher incidence of AEs in comparison to ICIs [2]. Thus, we should consider the balance of effectiveness/expecting prognosis and AEs.

In the present study, the median OS (13.4 months) after the initiation of SCT, as well as the ORR and DCR to SCT, was better in comparison to patients who received conventional treatment. In the phase III study (PFC) and phase II study (PC) the ORR was 36% and 54%, respectively [1,12]. The median PFS and OS of PC were 4.2 and 8.1 months, respectively, in patients receiving first-line therapy. Considering that SCT is administered as 2nd line therapy and beyond, this can be considered an excellent result, and it is considered that SCT after ICI should be performed as much as possible. Basic research also supports the superiority of immunotherapy followed by chemotherapy over chemotherapy, followed by immunotherapy [12]. Fridlender et al. [12] demonstrated that immunotherapy followed by chemotherapy, enhances the immune response to tumor cells and induces a therapeutic effect. In their experiments, mice bearing lung cancer tumors were intratumorally administered adenovirus-expressing interferon α (Ad–IFN-α) and then intravenously administered gemcitabine and cisplatin. This treatment sequence resulted in considerable tumor shrinkage, but only negligible tumor shrinkage was achieved with Ad-IFN-α alone, chemotherapy alone, and chemotherapy following Ad-IFN-α therapy. As in past reports, our data also showed that the treatment efficacy of PC was higher in comparison to S1, ORR was significantly superior, but not DCR. However, the superiority of the PC regimen to DCR, PFS, and OS was not statistically significant. On the other hand, regarding the incidence of Grade 3/4 AEs, patients treated with the PC regimen were more frequent than those treated with S1, careful monitoring should be recommended. Although the PC regimen as SCT might be more effective than S1, S1 may be preferable for some patients, such as those with relatively poor performance status.

Regarding the biomarker showing a therapeutic effect of SCT, we focused on CRP levels and NLR and at the start of SCT treatment, especially in the PC regimen. Systemic inflammation is recognized as a key determinant of the outcome in cancer patients [13]. Due to their widespread availability in clinical practice, CRP and the NLR were recognized as common markers that reflect systemic inflammation, and their relationship to the prognosis has been investigated in various malignancies. CRP levels have been reported to reflect the microenvironment and aggressiveness, such as tumor angiogenesis, in association with IL6 [15]. In addition, Neutrophils and lymphocyte have been used as parameters of the cancer-related inflammatory response [16,17]. Neutrophils are a type of inflammatory cell that secretes circulating vascular endothelial growth factor, chemokines, and proteases, which establish a tumor microenvironment through stimulating angiogenesis [17]. On the other hand, lymphocytes have been shown to be effective suppressors of cancer progression and to reflect host immunity [18]. And the NLR may be a more useful prognostic factor than the isolated leukocyte fraction as a marker reflecting both inflammation and the immune system function [4,17,18,19,20,21]. CRP and NLR have been studied as a prognostic biomarker of nivolumab in several cancers including NSCLC, metastatic renal cell carcinoma, metastatic melanoma, and R/M HNSCC [13,21,22,23]. Regarding SCT after ICI, NLR during nivolumab treatment was reported to increase over time in NSCLC, who were non-responders to SCT [4]. However, its utility in R/M HNSCCC patients treated with SCT has never been demonstrated. This study suggests that low pre-SCT CRP levels and NLR for a low pre-SCT NLR can predict a better prognosis in patients with R/M HNSCC after SCT.

The present retrospective study was associated with some limitations. First, CRP and NLR were identified as biomarkers based on the post hoc analysis, and this result needs to be interpreted with caution, due to multiplicity. Secondly, the study population was relatively small, which made it more difficult to perform a multivariate analysis. Thirdly, common HNSCCs, oral cancer, laryngeal cancer, hypopharyngeal cancer, and oropharyngeal cancer accounted for <70% of the cases. The others included paranasal SCCs and external auditory canal SCCs. These aspects should also be considered when interpreting the results. Prospective studies should be performed in the future.

## 4. Materials and Methods

### 4.1. Inclusion and Exclusion Criteria

All patients included in the study were 20 years of age or over, had a histologically confirmed diagnosis of R/M HNSCC not suitable for local definitive therapy. There had to be documented, measurable tumor, assessed by computed tomography or magnetic resonance imaging in two dimensions. The exclusion criteria included PS ≥ 3.

Nivolumab was administered to patients at a dose of 3 mg/kg or 240 mg/body every 2 weeks. Subsequently, PC or S1 was administered. Patients received weekly paclitaxel at a dose of 80 mg/m^2^, followed by weekly cetuximab at a dose of 400 mg/m^2^ on the first day and 250 mg/m^2^ on the other days. S1 is an orally active combination of tegafur (a prodrug that is converted by cells to fluorouracil), gimeracil (an inhibitor of dihydropyrimidine dehydrogenase, which degrades fluorouracil), and oteracil (an inhibitor of the phosphorylation of fluorouracil in the gastrointestinal tract, which thereby reduces the gastrointestinal toxic effects of fluorouracil) [24,25,26,27]. S1 was administered at a dose of 80–120 mg/body/day according to the body surface area in accordance with the drug information. S1 was administered for two weeks, with a one-week off period; this was repeated. Chemotherapies were selected depending on the status of each patient, which was determined by the head and neck cancer board in our institutions. Systemic comorbidities (e.g., renal and pulmonary dysfunction) were reviewed there. S1 treatment was chosen as SCT primarily because of a history of moderate or greater pulmonary dysfunction (emphysema and interstitial pneumonia). The patients received treatment until progression or the development of unacceptable toxicity. A biomarker analysis was only performed in the group of PC patients due to the limited number of cases.

### 4.2. Evaluation of the Response and Adverse Effects

The tumor response was evaluated using the Response Evaluation Criteria in Solid Tumors (version 1.1) based on the findings of computed tomography, which was performed every 8 to 12 weeks. PD was defined as a ≥20% increase in the sum of the diameters of the target lesions or the appearance of new metastatic lesions. SD was defined as ranging from a <30% decrease to a 20% increase in tumor size on imaging. A PR was defined as a ≥30% decrease in the sum of the diameters of target lesions. We evaluated the best overall response (BOR) of all patients as a CR, PR, SD, or PD. The ORR corresponded to a CR or PR, and the DCR corresponded to CR, PR, or SD. From the first day of treatment with SCT as the starting point, OS was assessed up until death, while PFS was assessed to the day of disease progression or death. Toxicity was assessed by the Common Terminology Criteria for Adverse Events (CTCAE) version 4.0. The study protocol was approved by the institutional review board of Kyushu University (reference number: 2020-77, 28 May 2020.). All patients gave their informed consent for participation in the study. This study was conducted in accordance with the principles of the Declaration of Helsinki.

### 4.3. Blood Data Evaluation

Serum CRP, the blood neutrophil count, and lymphocyte counts were obtained from records. Then, the NLR was calculated as the neutrophil count divided by the lymphocyte count. Pre-SCT CRP and pre-SCT NLR were defined as the serum CRP level and NLR within one week before the administration of PC, respectively.

### 4.4. Statistical Analyses

All calculations were performed using the JMP 14 software program (SAS Institute, Cary, NC, USA). The pre-SCT CRP and pre-SCT NLR were each divided into two groups (high or low values), and cutoff values were comprehensively determined by referring to a receiver operating characteristic curve and past reports. As a result, pre-SCT NLR was divided into the low group and the high group at the boundary of 7.0 and pre-SCT CRP at the boundary of 1.2 mg/dL, respectively. OS and PFS were calculated using the Kaplan-Meier method and were evaluated with the log-rank test, while categorical variables were analyzed using Fisher’s exact test. The risk was expressed as hazard ratios (HRs) and 95% confidence intervals (CIs). Univariate and multivariate Cox proportional hazards regression models were used to assess the associations between potential confounding variables and the PFS and OS. A multivariable analysis was performed after adjusting for sex, age, and Eastern Cooperative Oncology Group performance status (PS). *p* values of <0.05 were considered to indicate statistical significance.

## 5. Conclusions

In summary, this study elucidated the usefulness of the PC and S1 regimens as SCT after ICI for patients with R/M HNSCC. Furthermore, to our knowledge, this is the first study to show that a low pre-SCT CRP level and low pre-SCT NLR are good prognostic biomarkers for the PC regimen, when it is administered as SCT after ICI treatment for R/M HNSCC. These findings may help guide the choice of treatment and decision-making regarding whether to continue treatment, switch to another drug, or move to best-supportive-care-only treatment. Our findings can help improve the treatment strategy for palliative advanced-line chemotherapy for R/M HNSCC. The potential prognostic impact of these markers should be investigated further in a prospective study.

## Figures and Tables

**Figure 1 cancers-12-02299-f001:**
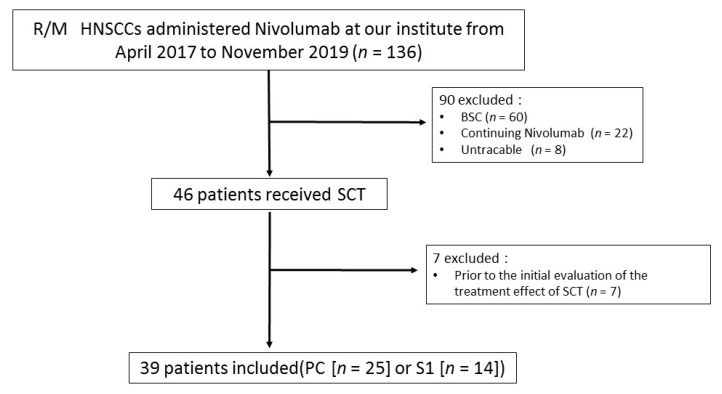
Flowchart of patient selection. R/M HNSCC, recurrent and/or head and neck squamous cell carcinoma. BSC, best supportive care; SCT, salvage chemotherapy; PC, combination chemotherapy of weekly paclitaxel and cetuximab; S1, tegafur-gimestat-otastat potassium.

**Figure 2 cancers-12-02299-f002:**
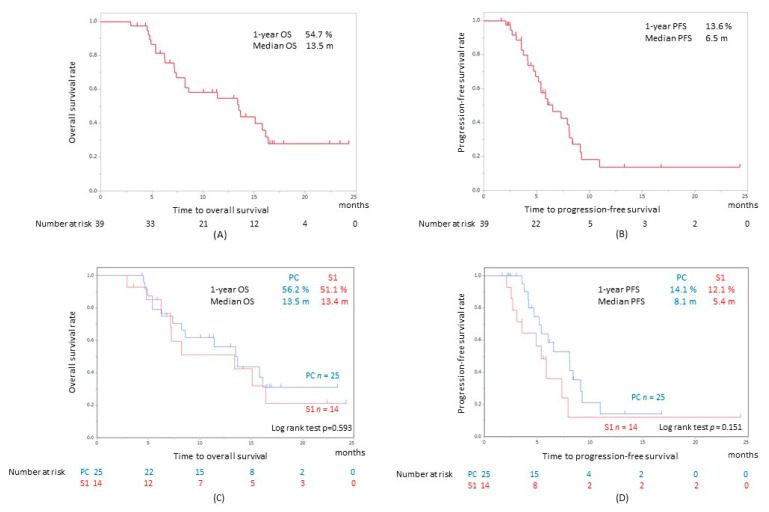
Kaplan-Meier curves for (**A**) OS and (**B**) PFS in all R/M HNSCC patients after SCT following nivolumab. The overall survival (**C**) and the progression free survival (**D**) between the patients administered PC and S1. OS, overall survival; PFS, progression free survival; PC, combination chemotherapy of weekly paclitaxel and cetuximab; S1, tegafur-gimestat-otastat potassium.

**Figure 3 cancers-12-02299-f003:**
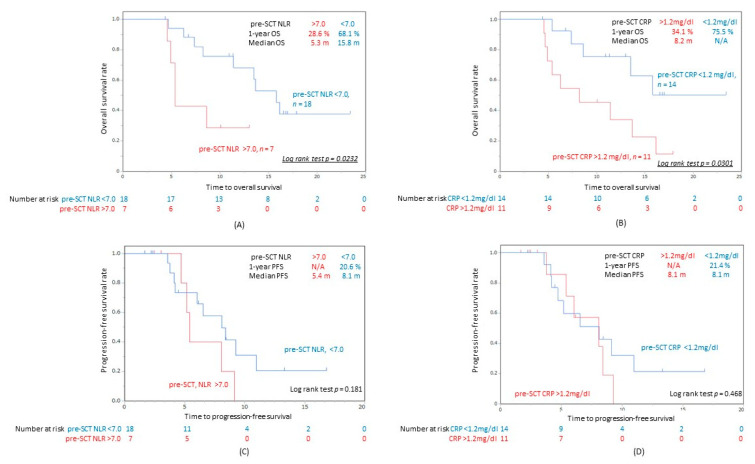
Kaplan-Meier curves for OS (**A**,**B**) and PFS (**C**,**D**) in R/M HNSCC patients after PC following the administration of nivolumab. (**A**) OS of patients with pre-SCT NLR > 7.0 and that of patients with pre-SCT NLR < 7.0. (**B**) OS of patients with pre-SCT CRP > 1.2 mg/dL and pre-SCT CRP < 1.2 mg/dL. (**C**) PFS of patients with pre-SCT NLR > 7.0 and that of patients with pre-SCT NLR < 7.0. (**D**) PFS of patients with pre-SCT CRP > 1.2 mg/dL and that of patients with pre-SCT CRP < 1.2 mg/dL. pre-SCT NLR, neutrophil to lymphocyte ratio at the initiation of SCT. CRP, serum C-reactive protein; pre-SCT CRP; CRP value at the initiation of SCT.

**Table 1 cancers-12-02299-t001:** Patient and tumor characteristics.

Characteristics	All (*n* = 39)		PC (*n* = 25)		S1 (*n* = 14)	
	*n*	%	*n*	%	*n*	%
Median Age		66 (33–78)		65 (33–77)		67 (40–73)
Age ≥65	22	56.4	14	56.0	8	57.1
Age <65	17	43.6	11	44.0	6	42.9
Gender						
Male	32	82.1	20	80.0	12	85.7
Female	7	17.9	5	20.0	2	14.3
Smoking status						
Brinkman index >1000	8	20.5	6	24.0	2	14.3
Brinkman index <1000	31	79.5	19	76.0	12	85.7
ECOG PS at the first administration of SCT						
PS 0–1	37	94.9	24	96.0	13	92.9
PS 2	2	5.1	1	4.0	1	7.1
Primary site						
Oral	9	23.1	7	28.0	2	14.3
Nasopharynx	4	10.3	2	8.0	2	14.3
Oropharynx	7	17.9	3	12.0	4	28.6
Hypopharynx	7	17.9	5	20.0	2	14.3
Larynx	2	5.1	2	8.0	0	0.0
Sinonasal tract	7	17.9	4	16.0	3	21.4
Others	3	7.7	2	8.0	1	7.1
Disease State						
LA	12	30.8	10	40.0	2	14.3
DM	11	28.2	7	28.0	4	28.6
LA + DM	16	41.0	8	32.0	8	57.1
Administration line of SCT in palliative therapy						
2nd	21	53.8	16	64.0	5	35.7
3rd	14	35.9	8	32.0	6	42.9
4th	3	7.7	1	4.0	2	14.3
5th	1	2.6	0	0.0	1	7.1
Cause of cessation of nivolumab						
PD	34	87.2	23	92.0	11	78.6
AE	4	10.3	1	4.0	3	21.4
PD + AE	1	2.6	1	4.0	0	0.0

ECOG PS, Eastern Cooperative Oncology Group performance status; LA, locally advanced disease; DM, distant metastasis; SCT, salvage chemotherapy; PD, progressive disease; AE, adverse event; PC, combination chemotherapy of weekly paclitaxel and cetuximab; S1, tegafur-gimestat-otastat potassium.

**Table 2 cancers-12-02299-t002:** The best response to salvage chemotherapy.

Response	All (*n* = 39)		PC (*n* = 25)		S1 (*n* = 14)		*p*-Value
	*n*	%	*n*	%	*n*	%	
CR	1	2.6	0	0	1	7.1	
PR	17	43.6	15	60	2	14.3	
SD	17	43.6	9	36	8	57.1	
PD	4	10.3	1	4	3	21.4	
ORR	18	46.2	15	60	3	21.4	0.0428
DCR	35	89.7	24	96	11	78.6	0.123

CR, complete response; PR, partial response; SD, stable disease; PD, progressive disease; ORR, objective response rate; DCR, disease control rate.

**Table 3 cancers-12-02299-t003:** The univariate analysis of factors associated with overall survival and progression-free survival in patients administered salvage chemotherapy with PC.

Factor		OS		PFS	
		HR (95% CI)	*p*	HR (95% CI)	*p*
Sex	Male/Female	0.0531 (0.00971–0.293)	0.0007	0.197 (0.0430–0.905)	0.0367
Age	<65/>65	0.599 (0.197–1.817)	0.365	0.467 (0.148–1.476)	0.195
Smoking status	B.I. <1000/>1000	0.663 (0.201–2.191)	0.500	0.566 (0.168–1.905)	0.358
SCT line	2nd (*n* = 16)/> 2nd (*n* = 9)	0.691 (0.240–1.989)	0.493	0.813 (0.288–2.300)	0.697
pre-SCT NLR	<7.0/>7.0	0.258 (0.0730–0.911)	0.0354	0.470 (0.151–1.458)	0.191
pre-SCT CRP	<1.2 mg/dL/>1.2 mg/dL	0.324 (0.0108–0.970)	0.0440	0.674 (0.231–1.969)	0.471

HR, hazard ratio, B.I., Brinkman Index.

**Table 4 cancers-12-02299-t004:** The multivariate analysis of factors associated with overall survival and progression-free survival in patients administered salvage chemotherapy with PC (reference to PS, Sex, and Age).

Factor		OS		PFS	
		HR (95% CI)	*p*	HR (95% CI)	*p*
pre-SCT NLR	<7.0/>7.0	0.498 (0.112–2.208)	0.359	0.839 (0.240–2.931)	0.783
pre-SCT CRP	<1.2 mg/dL/>1.2 mg/dL	0.0929 (0.0161–0.535)	0.0078	0.3411 (0.0897–1.298)	0.115

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
