# Peer review of "Prognostic Biomarkers of Salvage Chemotherapy Following Nivolumab Treatment for Recurrent and/or Metastatic Head and Neck Squamous Cell Carcinoma"

_cancers, 2020, doi:10.3390/cancers12082299_

Round 1

Reviewer 1 Report

Overview

This is a study that the authors investigated the outcome of salvage chemotherapy using paclitaxel/cetuximab or S-1 an oral fluoropyrimidine after becoming immune check point inhibitor refractory. The authors also mentioned about the prognostic factors affecting the patients’ survival and proved that a lower CRP/NLR before treatment significantly related to the patients’ survival in using paclitaxel and cetuximab.

Major Points

#1. In this manuscript, the primary conclusion was that a lower CRP/NLR significantly related to the patients survival in using paclitaxel and cetuximab as the salvage chemotherapy, because the title of the manuscript is “Prognostic biomarkers of salvage chemotherapy following nivolumab treatment for recurrent and/or metastatic head and neck squamous cell carcinoma”. This title is suitable for the special issue “CRP cancer” where the authors applied.

Then, I do not understand the reason why the authors included the patients treated with S-1 as subjects, who did not receive the biomarker analysis.

#2. The authors already reported the effect of the SCT following nivolumab therapy in another journal using small number of the patients (10 patients, reference #11). If some patients were common in two studies, the authors should declare about this issue or simply omit the data of used patients for possible meta-analysis in the future.

#3. The authors should make a brief comment about the treatment before ICI. Were all the patients platinum refractory? Were there any patients who received TPF or TPE before ICI?

This might influence on the selection of the SCT; PC or S-1.

#4. I do not understand the meaning of "administration line of SCT" in this manuscript. Did the authors try different regimens of chemotherapy before administrating PC/S-1 after becoming ICI refractory? If so, the authors should clarify the regimens of chemotherapy before PC/S-1 regimen.

Or simply counted all the treatment regimens in the whole period?

If so, was ICI counted as a line of chemotherapy?

This description was quite confusing.

Minor Points

P1L31:Abstract

“clarified the usefulness of the PC and S1”

In this abstract, S1 showed up suddenly here without any explanation. It is quite confusing. I recommend the authors to explain about the content of SCT earlier in the abstract.

P2L49-50

“To date, PC or S-1 was administered as SCT”

Explain abbreviations when they first appear in the manuscript.

There are 2 kinds of description demonstrating S1/S-1.

The authors should unify them.

Moreover, S-1 is not world-wide famous in the field of head and neck oncology however it is well known in the field of gastrointestinal oncology because of NEJM report in 2007. Even in the report of NEJM, S-1 was introduced as “S-1, an oral fluoropyrimidine” in the title of the report. I recommend the author to explain more about S-1 when it first appears in the manuscript.

P2L71

“In the 39 patients, the ORR and DCR of”

Explain abbreviations when they first appear in the manuscript.

P2L73

“The PS was 0-1 in 40 patients and”

Explain abbreviations when they first appear in the manuscript.

P2L89

“The estimated median OS and PFS”

Explain abbreviations when they first appear in the manuscript.

P3L89-90

“The estimated median OS and PFS from the first dose after SCT to the initiation of nivolumab therapy was 13.4 months and 6.5 months”

I am sorry I cannot understand what the authors meant.

Did the authors explain about the period between the initiation of nivolumab therapy and the first administration of their SCT?

I also want to know when the authors counted as the 1st day in estimating survival period, the first cycle of SCT or the last day of nivolumab administration.

P7L205

“receiving BSC only after progressive disease (PD)”

Explain abbreviations when they first appear in the manuscript

Author Response

Response to Reviewer 1 Comments

Major Points

#1. In this manuscript, the primary conclusion was that a lower CRP/NLR significantly related to the patients survival in using paclitaxel and cetuximab as the salvage chemotherapy, because the title of the manuscript is “Prognostic biomarkers of salvage chemotherapy following nivolumab treatment for recurrent and/or metastatic head and neck squamous cell carcinoma”. This title is suitable for the special issue “CRP cancer” where the authors applied.

Then, I do not understand the reason why the authors included the patients treated with S-1 as subjects, who did not receive the biomarker analysis.

Response:

Thank you for your pointing this out.

At our facility, we use two types of chemotherapy as SCT: PC and S1. In some cases, patients who cannot receive PC due to pulmonary dysfunction or other reasons may be able to receive S1. There are few reports on the efficacy of S1 for SCT. Although the cases were rare, the studies reported the therapeutic effect of S1. Unfortunately, due to the limited number of cases treated with S1, it was difficult to perform a statistical analysis of biomarkers. We therefore limited the statistical analysis of biomarkers to the cases treated with PC.

#2. The authors already reported the effect of the SCT following nivolumab therapy in another journal using small number of the patients (10 patients, reference #11). If some patients were common in two studies, the authors should declare about this issue or simply omit the data of used patients for possible meta-analysis in the future.

Response:

Thank you for your comment.

We discuss this issue in line 75. The 39 patients include 10 patients we have previously reported as a case series [11]. (line 75)

#3. The authors should make a brief comment about the treatment before ICI. Were all the patients platinum refractory? Were there any patients who received TPF or TPE before ICI?

This might influence on the selection of the SCT; PC or S-1.

Response:

Thank you for your comment.

We added a sentence about the treatment before ICI.

The previous palliative chemotherapies included platinum (16 of 39 patients [PC group, 8/25; S1 group, 8/14]), taxanes (5 of 39 patients [PC group, 2/25; S1 group, 3/14]) and cetuximab (17 of 39 patients [PC group, 8/25; S1 group, 9/14]). Thirty-one of 39 patients were platinum-refractory, and the others were platinum intolerant (n=8). (line81, Results)

#4. I do not understand the meaning of "administration line of SCT" in this manuscript. Did the authors try different regimens of chemotherapy before administrating PC/S-1 after becoming ICI refractory? If so, the authors should clarify the regimens of chemotherapy before PC/S-1 regimen.

Or simply counted all the treatment regimens in the whole period?

If so, was ICI counted as a line of chemotherapy?

This description was quite confusing.

Response:

Thank you for your comment.

We add the explanation about the lines of administration of SCT in line 92.

The line of administration of SCT was defined as the line of administration as palliative chemotherapy, including ICI therapy. Thus, we did not administer different chemotherapy regimens before administering PC/S1 after the patient became ICI-refractory. (line 92, Results)

Minor Points

P1L31:Abstract

“clarified the usefulness of the PC and S1”

In this abstract, S1 showed up suddenly here without any explanation. It is quite confusing. I recommend the authors to explain about the content of SCT earlier in the abstract.

 Response:

Thank you for your comment.

We have revised the Abstract according to your suggestion, and we explained the content of SCT earlier in the abstract. (line 24)

P2L49-50

“To date, PC or S-1 was administered as SCT”

Explain abbreviations when they first appear in the manuscript.

There are 2 kinds of description demonstrating S1/S-1.

The authors should unify them.

Moreover, S-1 is not world-wide famous in the field of head and neck oncology however it is well known in the field of gastrointestinal oncology because of NEJM report in 2007. Even in the report of NEJM, S-1 was introduced as “S-1, an oral fluoropyrimidine” in the title of the report. I recommend the author to explain more about S-1 when it first appears in the manuscript.

Response:

Thank you for your comment.

We added a sentence explaining about S-1 in line 225 of the Materials and Methods. (line 225)

P2L71

“In the 39 patients, the ORR and DCR of”

Explain abbreviations when they first appear in the manuscript.

Response:

Thank you for your comment.

We changed the sentence according to your comment. (line 87)

P2L73

“The PS was 0-1 in 40 patients and”

Explain abbreviations when they first appear in the manuscript.

Response:

Thank you for your comment. We changed the sentence according to the comment. (line 91)

P2L89

“The estimated median OS and PFS”

Explain abbreviations when they first appear in the manuscript.

 Response:

Thank you for your comment. We changed the sentence according to your comment.(line26)

P3L89-90

“The estimated median OS and PFS from the first dose after SCT to the initiation of nivolumab therapy was 13.4 months and 6.5 months”

I am sorry I cannot understand what the authors meant.

Did the authors explain about the period between the initiation of nivolumab therapy and the first administration of their SCT?

I also want to know when the authors counted as the 1st day in estimating survival period, the first cycle of SCT or the last day of nivolumab administration.

Response:

Thank you for your comment. We apologize for this mistake. The sentence was incorrect; it shows the duration of OS and PFS from the first day of SCT. It simply shows the OS and PFS of patients who received SCT.

P7L205

“receiving BSC only after progressive disease (PD)”

Explain abbreviations when they first appear in the manuscript

Response:

Thank you for your comment. We changed the sentence according to the comment. (line72)

Reviewer 2 Report

The manuscript deals with the actual and challenging issue of the recurrent and/or metastatic disease in head and neck cancer reporting a small mono-institutional experience on the use of salvage chemotherapy with paclitaxel and cetuximab in patients who underwent progression after a first line therapy using immunocheckpoints inhibitors. Also, they reported a possible prognostic role of the level of C-reactive protein and NLR ratio in these population. The manuscript might have interesting cues but need a profound editing before it can be considered.

Following, a first round of several major concerns that authors are invited to provide.

Major concerns

  • Authors are strongly invited to follow the standard structure of a scientific article (Introduction; Materials and Methods; Results; Discussion; Conclusions)
  • As reported by authors in the Introduction, the EXTREME Trial and the CheckMate 141 trial are the cornerstone of treatment for R/M HN cancers. The Extreme trial used the combination of Platinum 5 Fluororacil and Cetuximab in patients undergone to progression after first line treatment with curative intent whereas CheckMate 141 used Nivolumab in platinum- refractory cancers. Authors are invited to clarify the use of ICI as first line therapy in R/M HNC in their clinical practice and the subsequent use of salvage CT after progression. Also, the use of Paclitaxel and Cetuximab as well as the use of S1 should be clarified and supported by literature evidences. Are all considered patients platinum refractory?  If so, it should be better defined in the inclusion criteria (see the following point) together with a clear definition of “platinum refractory”. Otherwise, authors are invited to justify their choice not to use a platinum based CT in this patient population.Maybe some of the above aspects are already present in the text but need to come out more clearly in the text.
  • In Material and methods section a clear statements of inclusion and exclusion criteria for the enrollment in the study should be reported as well as a clear statement of the retrospective nature of the study in the Introduction. The subsection “Patients” (row 201-208) of Materials and Methods should be shifted to the subsection “Patients characteristics” of Results Section. Also, this sub-section is not clear to me. You state in the text that 87 patients (the total count is 90 as reported in the figure 3) were excluded! To SCT? If so it means they underwent to PD. Why 22 patients continued ICI? What do you mean 8 patients dropout? Drop out patients are in prospective and not in retrospective study! Moreover, among the 46 remaining patients you declare that 7 were excluded because the efficiency of SCT was never evaluated. What do you mean?
  • The last paragraph of Discussion section (row 191-198) and the paragraph of Conclusion are identical. Please remove one of them!
  • Results section: In general Figures must contain a brief and concise title and eventually captions to detail abbreviations used (i.e. SCT). Therefore, informations reported in the row 99-102 (Figure 1) and 127-135 (Figure 2) respectively, should be reported in the text if not already present or removed. Remove duplicate informations in tables and figures. I mean for example “pre-SCT NLR<7 n=18” is also reported in Table 3. Differences of two groups in the figures should be reported with different colors.
  • The discussion section is too short and really vague. Authors should discuss and argue the choice of using ICI as first line therapy in R/M HNC supporting it by reporting literature evidences in a clear manner. Row 149-151: “SCT is conventional chemotherapy”….what does it mean? SCT is the acronym of salvage chemotherapy thus salvage chemotherapy is conventional chemotherapy? “ recognizing the effectiveness……is an important” what? an important clinical issue? The overall limitations related to the retrospective nature of the study should be more comprensively argued. A wide subsection should be devoted to the limitations of the findings regarding the use of CRP and NLR as biomarkers as they come from a post-hoc analysis of retrospective data.

Author Response

Response to Reviewer 2 Comments

  • Authors are strongly invited to follow the standard structure of a scientific article (Introduction; Materials and Methods; Results; Discussion; Conclusions)
  • As reported by authors in the Introduction, the EXTREME Trial and the CheckMate 141 trial are the cornerstone of treatment for R/M HN cancers. The Extreme trial used the combination of Platinum 5 Fluororacil and Cetuximab in patients undergone to progression after first line treatment with curative intent whereas CheckMate 141 used Nivolumab in platinum- refractory cancers.
  • Authors are invited to clarify the use of ICI as first line therapy in R/M HNC in their clinical practice and the subsequent use of salvage CT after progression. Also, the use of Paclitaxel and Cetuximab as well as the use of S1 should be clarified and supported by literature evidences. Are all considered patients platinum refractory?  If so, it should be better defined in the inclusion criteria (see the following point) together with a clear definition of “platinum refractory”. Otherwise, authors are invited to justify their choice not to use a platinum based CT in this patient population. Maybe some of the above aspects are already present in the text but need to come out more clearly in the text.
  • The discussion section is too short and really vague. Authors should discuss and argue the choice of using ICI as first line therapy in R/M HNC supporting it by reporting literature evidences in a clear manner.

Response:

Thank you for your comment.

We added the following sentences in line 84, result section.

Thirty-one of 39 patients were platinum-refractory, and the others were platinum intolerant (n=8). Platinum refractory was defined as cancer progression within 6 months after the initiation of platinum chemotherapy in the context of primary or recurrent disease. (line81-, Result)

We also added the following sentences about the clinical usage of nivolumab in R/M HNSCC in line 166.

The National Comprehensive Cancer Network guideline describes nivolumab as a preferred regimen in cases with disease progression on or after platinum therapy. In our institutions, nivolumab is mainly administered for progressive disease after chemoradiotherapy (combined with platinum) or PFC therapy for R/M HNSCC. In the treatment of R/M HNSCC in the clinical setting, ICIs, including nivolumab, have been predominantly administered in early lines of treatment. (line 166, Discussion).

  • In Material and methods section a clear statements of inclusion and exclusion criteria for the enrollment in the study should be reported as well as a clear statement of the retrospective nature of the study in the Introduction. The subsection “Patients” (row 201-208) of Materials and Methods should be shifted to the subsection “Patients characteristics” of Results Section. Also, this sub-section is not clear to me. You state in the text that 87 patients (the total count is 90 as reported in the figure 3) were excluded! To SCT? If so it means they underwent to PD. Why 22 patients continued ICI? What do you mean 8 patients dropout? Drop out patients are in prospective and not in retrospective study! Moreover, among the 46 remaining patients you declare that 7 were excluded because the efficiency of SCT was never evaluated. What do you mean?

Response:

Thank you for your comment.

We moved the “Patients” subsection (row 201-208) of the Materials and Methods to the “Patients characteristics” subsection of the Results. The explanation of the patient's flow chart (Figure 3.→Figure 1.) is now more clearly stated, and some corrections were made (line 68, Patients characteristics in Result section). For example, we changed the word ‘dropout’ to ‘untraceable’

  • The last paragraph of Discussion section (row 191-198) and the paragraph of Conclusion are identical. Please remove one of them!

Response:

Thank you for your comment.

We removed the last paragraph of the Discussion.

  • Results section: In general Figures must contain a brief and concise title and eventually captions to detail abbreviations used (i.e. SCT). Therefore, informations reported in the row 99-102 (Figure 1) and 127-135 (Figure 2) respectively, should be reported in the text if not already present or removed. Remove duplicate informations in tables and figures. I mean for example “pre-SCT NLR<7 n=18” is also reported in Table 3. Differences of two groups in the figures should be reported with different colors.

Response:

Thank you for your comment.

We changed the tables and figures according to the comments, and removed duplicate information. We added the information about Figure 2 and 3 in the text.

Row 149-151: “SCT is conventional chemotherapy”….what does it mean? SCT is the acronym of salvage chemotherapy thus salvage chemotherapy is conventional chemotherapy? “ recognizing the effectiveness……is an important” what? an important clinical issue? The overall limitations related to the retrospective nature of the study should be more comprensively argued.

Thank you for your comment.

A wide subsection should be devoted to the limitations of the findings regarding the use of CRP and NLR as biomarkers as they come from a post-hoc analysis of retrospective data.

Response:

Thank you for your comment.

We changed the paragraph about the limitations. The main addition was the limitation regarding the post-hoc analysis. (line 211, Discussion)

Reviewer 3 Report

Dear authors,

I thank you for your interesting manuscript. here under my comments.

  1. Line 61 and further in text: please use S-1 or S1 uniform in whole text.
  2. Line 68: please change “oral region” to “oral cavity”
  3. Line 71; please define ORR and DCR here (Objective response rate and disease control rate)
  4. Line 110-111: Would you please explain different between skin disorders and grade 3 rash in your data
  5. Line 115: please explain more about timing: what was the timing? what was interval between ICI and SCT? what was interval between ICI and your test for NLR and CRP?
  6. Line 123: please report only HR in text and the rest in table.

Author Response

Response to Reviewer 3 Comments

  1. Line 61 and further in text: please use S-1 or S1 uniform in whole text.

Response:

Thank your pointing this out. We have used S1 uniformly.

  1. Line 68: please change “oral region” to “oral cavity”

Response:

Thank your pointing this out. This problem has been addressed. (line 79) 

  1. Line 71; please define ORR and DCR here (Objective response rate and disease control rate)

Response:

Thank your pointing this out. This problem has been addressed. (line 87)

  1. Line 110-111: Would you please explain different between skin disorders and grade 3 rash in your data

Response:

Thank your pointing this out. This problem has been addressed. This was our mistake. We changed ‘rash’ to skin disorder’. (line133, Result)

  1. Line 115: please explain more about timing: what was the timing? what was interval between ICI and SCT? what was interval between ICI and your test for NLR and CRP?

Response:

Thank your pointing this out. We added the following sentence.

The median interval between the last dose of nivolumab and the first dose of SCT in all patients was 21 days (14–394 days). (line140, 2.1.4. Association of pre-SCT NLR or pre-SCT CRP and the prognosis in patients treated with PC)

Pre-SCT CRP and pre-SCT NLR were defined as the serum CRP level and NLR within one week before the administration of PC, respectively. (line 252, Materials and Methods)

  1. Line 123: please report only HR in text and the rest in table.

Response: Thank your pointing this out. We have addressed this problem. (line144, Result)

Round 2

Reviewer 1 Report

Major Points

P3L91

The authors described here

“The line of administration of SCT was defined as the line of administration as palliative chemotherapy, including ICI therapy. The rates of second-line, third line, and fourth-line treatment were 53.8% (21/39), 35.9% (14/39), and 7.7% (3/39), respectively.”

This meant that 21 patients received only ICI and following SCT as palliative therapy, 14 patients received another line of chemotherapy before ICI and 4 patients received 2-3 lines of chemotherapy before ICI as palliative therapy.

And in the other part (P2L81-P3L84), the authors described “The previous palliative chemotherapies included platinum (16 of 39 patients [PC group, 8/25; S1 group, 8/14]), taxanes (5 of 39 patients [PC group, 2/25; S1 group, 3/14]) and cetuximab (17 of 39 patients [PC group, 8/25; S1 group, 9/14]).”

 Now we can understand that the majority of previous palliative chemotherapy was a combination of platinum and cetuximab.

These explanations should be in this order.

And please make them more understandable.

The authors can utilize another supplementary file to explain the definitive therapy and the lines of previous palliative chemotherapy of all 39 patients in this study.

Minor Points

P1L25:Abstract

“received S1 and oral fluoropyrimidine”

This should be “received S1, an oral fluoropyrimidine”

P2L49-50

“Ninety patients were excluded for the following reasons”

We do not usually use the word “exclude” before we define the inclusion criteria.

I recommend this part of explanation in a different way, such as followings:

From April 1, 2017 to November 30, 2019, 136 patients with R/M HNSCC were treated with nivolumab at Kyushu University Hospital, National Kyushu Cancer Center, and Kitakyushu Municipal Medical Center, Japan. All tumors were histologically confirmed to be squamous cell carcinoma. Twenty two patients were continuing ICI therapy because good responses were maintained. Sixty patients were selected to receive best supportive care (BSC) after progressive disease (PD) with ICI therapy. Eight patients became untraceable after ICI therapy. Finally, 46 patients were administered PC or S-1 as SCT after ICI therapy turned out to be PD and they were enrolled in this study.

P3L84

Thirty-one of 39 patients were platinum-refractory, and the others were platinum intolerant (n=8)”

This explanation was also indeterminate.

The word “platinum-refractory” is ordinary used for the reason why we administer nivolumab to the patients.

I would explain this issue as following.

Thirty-one of 39 patients were judged as platinum-refractory before administration of nivolumab because we experienced cancer progression within 6 months after the initiation of platinum chemotherapy in the context of primary or recurrent disease. Eight patients were judged as platinum intolerant before administration of nivolumab because of blah blah.

P8L169

“IN the treatment of R/M HNSCC in the clinical setting,” should be

“In the treatment of R/M HNSCC in the clinical setting,”

P8L174

“Thus, the recognition of the effectiveness and the AEs of SCT and the prediction of the prognosis after SCT are important clinical issues.”

 Does this mean “We should consider the balance of effectiveness/expecting prognosis and adverse event”?

P8L182

“Basic research also supports the superiority of”

Several words are in blue font and underlined.

Author Response

Response to Reviewer 1 Comments

â– Major Points

〇P3L91

The authors described here

“The line of administration of SCT was defined as the line of administration as palliative chemotherapy, including ICI therapy. The rates of second-line, third line, and fourth-line treatment were 53.8% (21/39), 35.9% (14/39), and 7.7% (3/39), respectively.”

This meant that 21 patients received only ICI and following SCT as palliative therapy, 14 patients received another line of chemotherapy before ICI and 4 patients received 2-3 lines of chemotherapy before ICI as palliative therapy.

And in the other part (P2L81-P3L84), the authors described “The previous palliative chemotherapies included platinum (16 of 39 patients [PC group, 8/25; S1 group, 8/14]), taxanes (5 of 39 patients [PC group, 2/25; S1 group, 3/14]) and cetuximab (17 of 39 patients [PC group, 8/25; S1 group, 9/14]).”

 Now we can understand that the majority of previous palliative chemotherapy was a combination of platinum and cetuximab. These explanations should be in this order.

Response:

Thank you for your comment.

We moved the following sentences to line 86, P2. 

‘The Eastern Cooperative Oncology Group performance status (PS)was 0–1 in 40 patients and 2 in two patients. They were treated with PC (25/39, 64.1%) or S1 (14/39, 35.9%) as SCT. The line of administration of SCT was defined as the line of administration as palliative chemotherapy, including ICI therapy. The rates of second-line, third line, and fourth-line treatment were 53.8% (21/39), 35.9% (14/39), and 7.7% (3/39), respectively. The previous palliative chemotherapies included platinum (16 of 39 patients [PC group, 8/25; S1 group, 8/14]), taxanes (5 of 39 patients [PC group, 2/25; S1 group, 3/14]) and cetuximab (17 of 39 patients [PC group, 8/25; S1 group, 9/14]). ‘ 

〇And please make them more understandable.

The authors can utilize another supplementary file to explain the definitive therapy and the lines of previous palliative chemotherapy of all 39 patients in this study.

Response:

Thank you for your comment.

We added Table 2S, Summary of the previous definitive therapies and the administration lines of SCT, and the sentence in the manuscript. (Line88, P3)

â– Minor Points

〇P1L25:Abstract

“received S1 and oral fluoropyrimidine”

This should be “received S1, an oral fluoropyrimidine”

Response: Thank you for your comment. We changed the sentence. (Line 25, P1, Abstract)

〇P2L49-50

“Ninety patients were excluded for the following reasons”

We do not usually use the word “exclude” before we define the inclusion criteria.

I recommend this part of explanation in a different way, such as followings:

From April 1, 2017 to November 30, 2019, 136 patients with R/M HNSCC were treated with nivolumab at Kyushu University Hospital, National Kyushu Cancer Center, and Kitakyushu Municipal Medical Center, Japan. All tumors were histologically confirmed to be squamous cell carcinoma. Twenty-two patients were continuing ICI therapy because good responses were maintained. Sixty patients were selected to receive best supportive care (BSC) after progressive disease (PD) with ICI therapy. Eight patients became untraceable after ICI therapy. Finally, 46 patients were administered PC or S-1 as SCT after ICI therapy turned out to be PD and they were enrolled in this study.

Response: Thank you for your comment. We changed the sentence. (Line70 ,P2, Results)

〇P3L84

Thirty-one of 39 patients were platinum-refractory, and the others were platinum intolerant (n=8)”

 This explanation was also indeterminate.

The word “platinum-refractory” is ordinary used for the reason why we administer nivolumab to the patients.

I would explain this issue as following.

Thirty-one of 39 patients were judged as platinum-refractory before administration of nivolumab because we experienced cancer progression within 6 months after the initiation of platinum chemotherapy in the context of primary or recurrent disease. Eight patients were judged as platinum intolerant before administration of nivolumab because of renal dysfunction, old age, or poor PS.

Response: Thank you for your comment. We changed the sentence. (Line94, P3, Results)

〇P8L169

“IN the treatment of R/M HNSCC in the clinical setting,” should be

“In the treatment of R/M HNSCC in the clinical setting,”

Response: Thank you for your comment. We changed the sentence. (Line 179, P8, Discussion)

〇P8L174

“Thus, the recognition of the effectiveness and the AEs of SCT and the prediction of the prognosis after SCT are important clinical issues.”

 Does this mean “We should consider the balance of effectiveness/expecting prognosis and adverse event”?

Response: Thank you for your comment. We changed the sentence. (Line 185, P8, Discussion)

〇P8L182

“Basic research also supports the superiority of”

Several words are in blue font and underlined.

Response: Thank you for your comment. We changed the sentence. (Line192, P8, Discussion)

Reviewer 2 Report

Comments

I would like to thank the authors for the editings. The paper is more fluent and readable now. Anyway some points need to be further addressed before the manuscript can be published.

1) Authors are strongly invited to follow the standard structure of a scientific article (Introduction; Materials and Methods; Results; Discussion; Conclusions)

The structure of the manuscript has not been modified according to the standard format (Introduction; Materials and Methods; Results; Discussion; Conclusions). It is not acceptable to set the manuscript with Materials and Methods at the end of the article. The standard format is above indicated as well as in the “Authors’ guidelines” of the journal. 

2) I still not find a reference literature to support the use of tegafur-gimestat-otastat potassium S1 in recurrent HNC . The reference 25 (Lines 224-227) is related to the use of it in Gastric cancer and not in Head and Neck cancers. Please provide a reference of literature to support it! Otherwise explain and argument your decision of using it in clinical practice.

Also, a section on methodology should be provided in materials and methods section reporting inclusion and exclusion general criteria of enrollment, criteria for patients candidates to PC or S1, Performance status, age  etc…Also, the platinum refractory definition, that maybe is an inclusion criteria for the study, should be reported in this section.

3) Results

Lines 74-75 I can’t understand why 7 patients were excluded. “Seven patients were excluded prior to the initial evaluation of the treatment effect of SCT. What does it mean? They interrupted the treatment?  Please explain it in the text!

When you indicate the number of patients use only numbers or only letters according to authors’ guidelines (136 patients with R/M-HNSCC…….Ninety patients……46 patients……seven patients etc)  

4) Discussion

Lines 182-184 “Basic research……followed by immunotherapy” This is an interesting point that could value the importance of your study. Please argument the rational of using ICI before chemotherapy rather than the opposite.

Author Response

Response to Reviewer 2 Comments

1) Authors are strongly invited to follow the standard structure of a scientific article (Introduction; Materials and Methods; Results; Discussion; Conclusions)

The structure of the manuscript has not been modified according to the standard format (Introduction; Materials and Methods; Results; Discussion; Conclusions). It is not acceptable to set the manuscript with Materials and Methods at the end of the article. The standard format is above indicated as well as in the “Authors’ guidelines” of the journal.

Response:

We apologize for our failure to respond to this question last time.

We followed the authors guideline of Cancers and used their template to create the manuscript.

2) I still not find a reference literature to support the use of tegafur-gimestat-otastat potassium S1 in recurrent HNC . The reference 25 (Lines 224-227) is related to the use of it in Gastric cancer and not in Head and Neck cancers. Please provide a reference of literature to support it!

Response:

We added the reference 24-26 about S1 treatment for head and neck cancers. Especially, new reference 24 and 25 represents about efficacy of R/M HNSCC. (Line243, P9, Materials and Methods) 

Also, a section on methodology should be provided in materials and methods section reporting inclusion and exclusion general criteria of enrollment, criteria for patients candidates to PC or S1, Performance status, age etc…Also, the platinum refractory definition, that maybe is an inclusion criteria for the study, should be reported in this section.

Response:

â‘ We added section of ‘Inclusion and Exclusion Criteria’ in Materials and Methods.

All patients included in the study were 18 years of age or over, had a histologically confirmed diagnosis of R/M HNSCC not suitable for local definitive therapy. There had to be documented, measurable tumor, assessed by computed tomography or magnetic resonance imaging in two dimensions. Exclusion criteria included PS>3 and development of advanced second primary tumors. (Line236, P9, Materials and Methods)

â‘¡We changed the sentences in Line235, P9.

‘Chemotherapies were selected depending on the status of each patient, which was determined by the head and neck cancer board in our institutions. Systemic comorbidities (e.g., renal and pulmonary dysfunction) were reviewed there. S1 treatment was chosen as SCT primarily because of a history of moderate or greater pulmonary dysfunction (emphysema and interstitial pneumonia). ‘

â‘¢We described about the platinum refractory definition according to reviewer 1’s comment. (Line98, P3, Results)

3) Results

Lines 74-75 I can’t understand why 7 patients were excluded. “Seven patients were excluded prior to the initial evaluation of the treatment effect of SCT. What does it mean? They interrupted the treatment?  Please explain it in the text!

Response:

Thank you for your comment. We changed the sentences.

Seven patients were not included this study because they were before the first assessment of treatment efficacy after the start of SCT. (Line77, P2, Results)

When you indicate the number of patients use only numbers or only letters according to authors’ guidelines (136 patients with R/M-HNSCC…….Ninety patients……46 patients……seven patients etc) 

Response:

Thank you for your comment.

We rechecked the numbers according to ‘6. 1 Numbers’ in Author’s guideline.

It said that, with a few exceptions, numbers should usually be written as numbers; numbers from 0 to 9 should be written as words, except when accompanied by units. It also said that if a sentence begins with a number, the numbers should always be written out in full.

4) Discussion

Lines 182-184 “Basic research……followed by immunotherapy” This is an interesting point that could value the importance of your study. Please argument the rational of using ICI before chemotherapy rather than the opposite.

Response:

Thank you for your comment.

We added the following sentence.

Fridlender ZG et al demonstrated that immunotherapy followed by chemotherapy enhances the immune response to tumor cells and induces a therapeutic effect [13]. In their experiments, mice bearing lung cancer tumors were intratumorally administered adenovirus-expressing interferon α (Ad–IFN-α) and then intravenously administered gemcitabine and cisplatin. This treatment sequence resulted in considerable tumor shrinkage, but only negligible tumor shrinkage was achieved with Ad-IFN-α alone, chemotherapy alone, and chemotherapy following Ad-IFN-α therapy. (Line194, P8, Results)
